

# Mechanical stability of resonant Bose-Fermi mixtures

Christian Gualerzi[1], Leonardo Pisani[2,3]* and Pierbiagio Pieri[2,3]†

**1** CSE - Consorzio Servizi Bancari, Via Emilia 272, I-40068, San Lazzaro di Savena (BO), Italy
**2** Dipartimento di Fisica e Astronomia "Augusto Righi",
Università di Bologna, Via Irnerio 46, I-40126, Bologna, Italy
**3** INFN, Sezione di Bologna, Viale Berti Pichat 6/2, I-40127, Bologna, Italy

★ leonardo.pisani2@unibo.it , † pierbiagio.pieri@unibo.it

## Abstract

We investigate the mechanical stability of Bose-Fermi mixtures at zero temperature in the presence of a tunable Feshbach resonance, which induces a competition between boson condensation and boson-fermion pairing when the boson density is smaller than the fermion density. Using a many-body diagrammatic approach validated by fixed-node Quantum Monte Carlo calculations and supported by recent experimental observations, we determine the minimal amount of boson-boson repulsion required to guarantee the stability of the mixture across the entire range of boson-fermion interactions from weak to strong coupling. Our stability phase diagrams indicate that mixtures with boson-to-fermion mass ratios near two, such as the $^{87}$Rb-$^{40}$K system, exhibit optimal stability conditions. Moreover, by applying our results to a recent experiment with a $^{23}$Na-$^{40}$K mixture, we find that the boson-boson repulsion was insufficient to ensure stability, suggesting that the experimental timescale was short enough to avoid mechanical collapse. On the other hand, we also show that even in the absence of boson-boson repulsion, Bose-Fermi mixtures become intrinsically stable beyond a certain coupling strength, preceding the quantum phase transition associated with the vanishing of the bosonic condensate. We thus propose an experimental protocol for observing this quantum phase transition in a mechanically stable configuration.

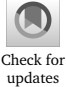

# 1   Introduction

Recently, mixtures of single-component fermions and bosons with a tunable Bose-Fermi (BF) interaction have been studied quite extensively in the field of ultra-cold gases, both experimentally [1–30] and theoretically [31–62].

A motivation for their interest is their possible use as quantum simulators of analogous systems arising in other contexts in physics. Examples are high-density quark matter, in which unpaired quarks (fermions) interact with diquarks (bosons) [63], or p-wave superfluids, which, according to different recent proposals [64–67], could be realized with BF mixtures in ultra-cold-atom platforms.

An alternative motivation is the possibility of exploring novel questions of intrinsic interest in quantum many-body physics. A notable example in this respect is the issue of competition between boson-fermion pairing and boson condensation when the BF interaction is made progressively stronger by means of a Feshbach resonance [31, 42, 45, 48, 52, 58].

For both narrow [31] and broad resonances [42, 45, 48, 52], it has been shown that, when the Feshbach resonance is adiabatically crossed coming from the weakly attractive side, the progressive build-up of BF pairing correlations increasingly depletes the boson condensate. For mixtures in which the boson density $n_B$ does not exceed the fermion density $n_F$, the condensate vanishes completely beyond a critical BF coupling strength even at zero temperature [31, 42, 45, 48, 52, 58]. In addition, Ref. [52] found a nearly universal behavior for the condensate fraction $n_0/n_B$, with $n_0/n_B$ depending essentially only on the BF coupling strength, regardless of the relative concentration $n_B/n_F$.

This prediction has been quantitatively confirmed in a recent experiment with a $^{23}$Na–$^{40}$K BF mixture. In such an experiment, the BF interaction was varied by sweeping a broad Feshbach resonance between Na and K atoms [25], while the interaction between Na atoms (which constituted the bosonic component of the mixture) was weakly repulsive. Previous theoretical works on BF mixtures indicate, however, that a too weak BB repulsion may not be sufficient to guarantee the mechanical stability of the BF mixture when crossing the whole BF Feshbach resonance. The question is then whether the BB repulsion in the experiment [25] was strong enough to guarantee mechanical stability. This question motivates the present work.

The stability of Bose-Fermi mixtures with respect to collapse or phase separation has already been analyzed in previous works. Early treatments did not consider the possibility of tuning the BF interaction with a Feshbach resonance, and therefore adopted approximations that assume a weak BF coupling strength [68–74]. A more recent work using diagrammatic methods [75] also assumed a weak BF interaction.

The first works considering the stability of a BF mixture in the presence of a resonant BF interaction that induces BF pairing focused on the case of a narrow Feshbach resonance [31, 40]. In this case, the corresponding "two-channel" Hamiltonian, which explicitly includes fermionic molecules on top of the bosonic and fermionic atoms, can easily be diagonalized. This is because, in the limit of a narrow resonance, bosonic operators can be replaced with condensate amplitudes in the interaction part of the Hamiltonian [31, 40].

The case of a broad resonance was instead first addressed in [43] with the use of a Jastrow-Slater variational wave function to describe the ground state of a resonant BF mixture. The evaluation of the variatonal energy was performed within the lowest-order constrained variational (LOCV) approximation, and a small concentration of bosons ($n_B/n_F \ll 1$) was assumed. However, the Jastrow-Slater variational wave function is not justified on the molecular side of the Feshbach resonance, when the binding of bosonic and fermionic atoms into fermionic molecules becomes relevant.

The works [56, 57] used instead a partially [56] or fully [57] self-consistent T-matrix approach to analyze the stability of a BF mixture with a broad Feshbach resonance. However, their analysis was restricted to the normal phase, that is, in the absence of a boson condensate, and cannot be straightforwardly extended to the condensed phase (of relevance to the experiment [25]) due to the well-known dichotomy between conserving and gapless approximations for condensed bosons [76].

Finally, recent work [60] tackled the problem with a diagrammatic approach for the condensed phase that explicitly includes fermion-mediated interactions in the bosonic self-energy. However, this approach was constructed for BF mixtures in which the condensate depletion is negligible (the feedback of noncondensed bosons on the fermion self-energy was altogether neglected). This is what might happen when the boson density greatly exceeds the fermion density and the (metastable) "repulsive branch" is considered for positive BF scattering lengths (see [60] and the Supplementary Material therein), but it is not relevant for the case corresponding to the experimental situation of [25], with $n_B < n_F$ and a condensate fraction that progressively decreases and eventually vanishes on the molecular side of the Feshbach resonance.

In the present work, we will address exactly this case. Building on the same many-body diagrammatic approach [52] validated by the experiment [25] and by fixed-node quantum Monte Carlo simulations [52] we analyze the mechanical stability of the mixture across the entire resonance sweep, and assess the minimal value of the BB repulsion required to avoid instabilities under a variety of conditions. Focusing on the experiment [25], we will see that the repulsion was more than an order of magnitude smaller than the minimum value required to ensure stability throughout the entire sweep from weak to strong BF attraction. This is rather surprising, since no mechanical collapse was observed during the experiment [25]. A possible explanation is that the timescale of the experiment was short enough to avoid mechanical collapse, as also argued in [25]. At the same time, we demonstrate an unexpected intrinsic stability that emerges before the condensate vanishes, suggesting a route for experimentally detecting the associated quantum phase transition in a mechanically stable configuration.

In our calculations, we will focus on homogeneous mixtures, without addressing the inhomogeneities introduced by a confining potential. On the one hand, in the experiment [25] the frequencies of the harmonic potentials acting on the two species were specifically tailored to have nearly-matched densities in the central region of the trap which, within a local density approximation, can be treated as locally uniform. On the other hand, techniques are now available to trap ultracold gases in essentially uniform potentials for both bosons [77] and fermions [78].

The paper is organized as follows. In Sec. 2 we describe the model Hamiltonian and the many-body diagrammatic theoretical approach used in the present work. This is mainly a recap of the formalism already used in [52]. However, we will also discuss an improvement on the way the BB repulsion is included by the formalism that is particularly important when stability is at issue. Section 3 presents our results for stability. In particular, we first discuss in Sec. 3.1 the numerical results for the thermodynamic quantities $\mu_B, \mu_F, n_0$ and their asymptotic behavior in both weak- and strong-coupling limits. These quantities are the necessary input for the calculation of the stability matrix, which is analyzed in Sec. 3.2. Based on this matrix,

we then construct in Sec. 3.3 the resulting stability phase diagram for different density and mass ratios between the two species. Sec. 4 reports our conclusions. Finally, the appendix reports second-order perturbative results in the strong-coupling limit of our approach.

# 2 Formalism

## 2.1 The system Hamiltonian

We consider a homogeneous mixture of spin-polarized fermions, with mass $m_F$ and number density $n_F$, interacting with single-component bosons, with mass $m_B$ and number density $n_B$, at zero temperature in three dimensions. The system is assumed to be dilute, such that the range of all interactions is smaller than the average inter-particle distance. The BB interaction is assumed to be short-ranged and weakly repulsive. The BF interaction is instead assumed to be tunable by a broad Fano-Feshbach resonance, for which the effective range of the attractive potential is much smaller than the corresponding scattering length. Finally, interactions between fermions can be neglected since $s$-wave interactions are forbidden by the Pauli exclusion principle, while higher angular momentum interactions are strongly suppressed for dilute gases.

Under these assumptions, the mixture can be described by a grand-canonical Hamiltonian of the form:

$$H = \sum_{s=B,F} \int d\mathbf{r}\, \psi_s^\dagger(\mathbf{r}) \left( -\frac{\nabla^2}{2m_s} - \mu_s \right) \psi_s(\mathbf{r}) + v_0^{BF} \int d\mathbf{r}\, \psi_B^\dagger(\mathbf{r}) \psi_F^\dagger(\mathbf{r}) \psi_F(\mathbf{r}) \psi_B(\mathbf{r})$$
$$+ \frac{1}{2} \int d\mathbf{r} \int d\mathbf{r}'\, \psi_B^\dagger(\mathbf{r}) \psi_B^\dagger(\mathbf{r}') U_{BB}(\mathbf{r}-\mathbf{r}') \psi_B(\mathbf{r}') \psi_B(\mathbf{r}), \tag{1}$$

where (for s = B, F) the field operators $\psi_s^\dagger(\mathbf{r})$ and $\psi_s(\mathbf{r})$ create and destroy, respectively, a particle of mass $m_s$ at position $\mathbf{r}$. We set $\hbar = k_B = 1$ throughout this paper.

The first term in Eq. (1) corresponds to the grand-canonical Hamiltonian for non-interacting BF mixtures, where $\mu_s$ is the chemical potential for s = B, F, while the second and third terms represent the BF and BB interactions. In particular, the tunable BF interaction is described by an attractive point-contact potential, whose bare coupling constant $v_0^{BF}$ is expressed in terms of the BF scattering length $a_{BF}$ with the same regularization procedure commonly used for Fermi gases [79–81] (see Eq. (4) below). The intensity of the BF interaction is then conveniently parametrized in terms of the dimensionless BF coupling strength $g_{BF} = (k_F a_{BF})^{-1}$ with $k_F = (6\pi^2 n_F)^{1/3}$. For weak BF attraction, $a_{BF}$ is small and negative (such that $g_{BF} \ll -1$) and perturbation theory is applicable [71,82]. For strong BF attraction, $a_{BF}$ is small and positive (such that $g_{BF} \gg 1$), the two-body problem binding energy $\epsilon_0 = 1/(2m_r a_{BF}^2)$ is large and the system effectively becomes a mixture of molecules and unpaired fermions [51] ($m_r = m_B m_F/(m_B + m_F)$ is the reduced mass). In this respect, we stress that we focus here on the case $n_B < n_F$, as relevant to the experiment [25] and for which a full competition between boson condensation and pairing of bosons with fermions is allowed.

The BB interaction $U_{BB}$ is also short-ranged but only weakly repulsive so that a simple perturbative approach can be employed: $U_{BB}(\mathbf{r}-\mathbf{r}')$ is replaced by an effective interaction of the form $\frac{4\pi a_{BB}}{m_B}\delta(\mathbf{r}-\mathbf{r}')$ in perturbative expressions (where $a_{BB}$ is the BB scattering length). For the BB interaction, we will use as dimensionless interaction parameter $\zeta_{BB} = k_F a_{BB}$. Note that the gas parameter $n_B a_{BB}^3 = x\zeta_{BB}^3/6\pi^2$, where $x = n_B/n_F$ is the boson concentration. Values of $\zeta_{BB} < 1$ will thus guarantee, for $x < 1$, $n_B a_{BB}^3 \lesssim 1.7 \times 10^{-2}$ and thus a fractional condensate depletion due to BB repulsion, $8/3(n_B a_{BB}^3/\pi)^{1/2}$ [83], smaller than about 0.2, i.e., within the region where the BB repulsion can reasonably be considered weak [84].

(a)
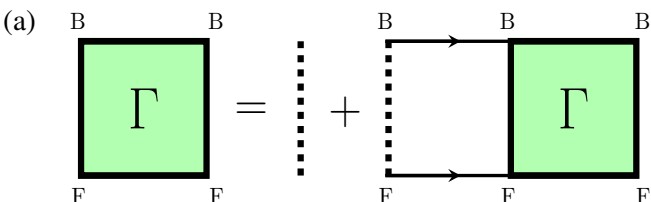

(b)
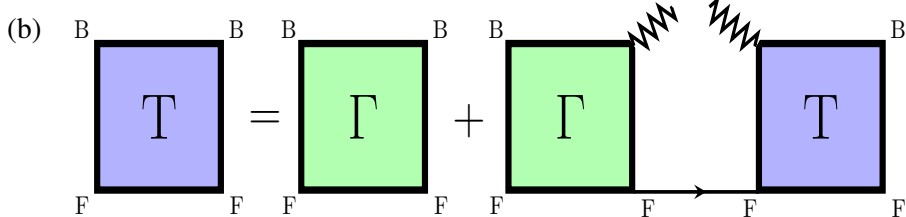

Figure 1: Diagrammatic representation of (a) the particle-particle ladder $\Gamma(P)$ and (b) the many-body $T$-matrix $T(P)$ in the condensed phase. Full lines correspond to bare boson (B) and fermion (F) Green's functions, dashed lines to the bare BF interaction of strength $v_0^{\mathrm{BF}}$ and zigzag lines to condensate factors $\sqrt{n_0}$.

It should be noted that in the experiment [25] the BB repulsion $a_{\mathrm{BB}}$ remained at its natural value for $^{23}$Na atoms ($a_{\mathrm{BB}} = 0.53 a_0$, where $a_0$ is the Bohr radius), which, with the values of the fermion density reported in [25] and considering $n_{\mathrm{B}} < n_{\mathrm{F}}$ as in the experiment, corresponds to $n_{\mathrm{B}} a_{\mathrm{BB}}^3 \lesssim 3 \times 10^{-7}$, well within the weak repulsion regime.

## 2.2 Diagrammatic theory

The present work is based on the diagrammatic approach adopted in [52], where a homogeneous BF mixture was studied in the condensed phase within a non-self-consistent $T$-matrix approximation. In the normal phase, pairing correlations between bosons and fermions are captured by the particle-particle ladder $\Gamma(\mathbf{P}, \Omega)$ shown in Fig. 1a, which describes an infinite series of repeated BF scattering events. It satisfies the Bethe-Salpeter equation

$$\Gamma(\mathbf{P}, \Omega) = v_0^{\mathrm{BF}} - v_0^{\mathrm{BF}} \Gamma(\mathbf{P}, \Omega) \int \frac{d\mathbf{p}}{(2\pi)^3} \int_{-\infty}^{\infty} \frac{d\omega}{2\pi} G_{\mathrm{F}}^0(\mathbf{P} - \mathbf{p}, \Omega - \omega) G_{\mathrm{B}}^0(\mathbf{p}, \omega), \tag{2}$$

where $G_{\mathrm{B}}^0, G_{\mathrm{F}}^0$ are bare boson and fermion Green's function

$$G_s^0(\mathbf{k}, \omega) = \frac{1}{i\omega - \xi_{\mathbf{p}}^s}, \qquad s = \mathrm{B, F}, \tag{3}$$

with $\xi_{\mathbf{p}}^s = p^2/2m_s - \mu_s$. In the present work, the zero-temperature limit is taken by considering the continuum limit of the Matsubara imaginary frequencies. By performing the frequency integral in Eq. (2) and using the regularization of the contact potential [79–81]

$$\frac{1}{v_0^{\mathrm{BF}}} = \frac{m_r}{2\pi a_{\mathrm{BF}}} - \int \frac{d\mathbf{k}}{(2\pi)^3} \frac{2m_r}{\mathbf{k}^2}, \tag{4}$$

one obtains

$$\Gamma(\mathbf{P}, \Omega)^{-1} = \frac{m_r}{2\pi a_{\mathrm{BF}}} - \frac{m_r^{\frac{3}{2}}}{\sqrt{2}\pi} \left[ \frac{P^2}{2M} - \mu_{\mathrm{F}} - \mu_{\mathrm{B}} - i\Omega \right]^{\frac{1}{2}} - \int \frac{d\mathbf{p}}{(2\pi)^3} \frac{\Theta\left(-\xi_{\mathbf{P}-\mathbf{p}}^{\mathrm{F}}\right)}{\xi_{\mathbf{P}-\mathbf{p}}^{\mathrm{F}} + \xi_{\mathbf{p}}^{\mathrm{B}} - i\Omega}, \tag{5}$$

where $M = m_{\mathrm{B}} + m_{\mathrm{F}}$ and $\Theta(x)$ is the Heaviside step function of argument $x$.

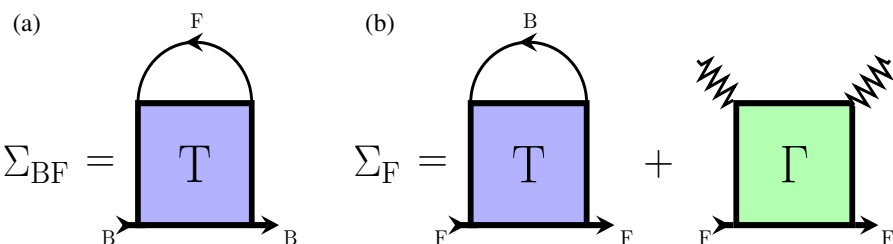

Figure 2: Feynman's diagrams for (a) the contribution to the boson self-energy $\Sigma_{BF}$ arising from interactions with fermions and (b) the fermion self-energy $\Sigma_F$. Full lines correspond to bare boson (B) and fermion (F) Green's functions, and zigzag lines to condensate factors $\sqrt{n_0}$.

Equation (4) is obtained by representing the BF attractive contact potential as the limit of a separable potential $V(k,k') = v_0^{BF}\Theta(k_0-k)\Theta(k_0-k')$ when the momentum range $k_0 \to \infty$, and the scattering length $a_{BF}$ is kept fixed. Specifically, when $k_0$ is finite, one obtains for the zero-energy limit of the on-shell $t$-matrix $t(k,k')$ of scattering theory [85]

$$t(0,0) = \left[ \frac{1}{v_0^{BF}} + \int \frac{d\mathbf{k}}{(2\pi)^3} \frac{2m_r}{\mathbf{k}^2}\Theta(k_0-k) \right]^{-1} , \tag{6}$$

which, together with the standard relation $t(0,0) = 2\pi a_{BF}/m_r$ connecting the on-shell $t$-matrix to the scattering length, yields the formal Eq. (4). Note that, before taking the limit $k_0 \to \infty$, a simple integration yields

$$v_0^{BF} = -\frac{\pi}{m_r k_0}\left(1 + \frac{2\pi^2}{a_{BF}k_0}\right) = -\frac{\pi}{m_r k_0}\left(1 + 2\pi^2 g_{BF}\frac{k_F}{k_0}\right). \tag{7}$$

The latter expression highlights that the finite range potential $V(k,k')$ is always attractive independently of the sign of $a_{BF}$ when $k_0$ is increased toward infinity (to represent a zero-range potential in real space).

The momentum integral in Eq. (5) can be performed analytically (see Eq. (16) of Ref. [46]). Note that Eq. (5) is derived under the assumption that $\mu_B \leq 0$, a condition that is always verified except in the weak coupling limit of the BF interaction, for strong enough BB repulsion. In this limit, the use of bare bosonic Green's functions with a positive $\mu_B$ when constructing the self-energies is unphysical since they yield a negative momentum distribution function in the region where $\xi_B(\mathbf{k})$ is negative. Physical results are recovered by using the unperturbed value $\mu_B = 0$ for a non-interacting Bose gas at $T = 0$ in the bare bosonic Green's function [86]. Therefore, whenever $\mu_B > 0$, we will set it to zero in Eq. (5) as well as in the bare bosonic Green's function appearing in Eq. (11) below for the fermionic self-energy.

In the condensed phase an additional event is to be considered, that of a boson being scattered into the condensate, as illustrated diagrammatically in Fig. 1b. The resulting Bethe-Salpeter equation for the $T$-matrix in the condensed phase $T(\mathbf{P},\Omega)$ thus reads

$$T(\mathbf{P},\Omega) = \Gamma(\mathbf{P},\Omega) + n_0\Gamma(\mathbf{P},\Omega)\,G_F^0(\mathbf{P},\Omega)\,T(\mathbf{P},\Omega) , \tag{8}$$

with $n_0$ the condensate density. One immediately obtains

$$T(\mathbf{P},\Omega) = \frac{1}{\Gamma(\mathbf{P},\Omega)^{-1} - n_0 G_F^0(\mathbf{P},\Omega)} . \tag{9}$$

The ensuing bosonic and fermionic self-energies are shown in Fig. 2, respectively, and are obtained by closing the $T$-matrix in the condensed phase with a free fermion or boson propagator. In the fermionic case an additional contribution stems from the interaction with bosons in the condensate (zigzag lines in Fig. 2). In this case, however, the many-body $T$-matrix in the condensed phase $T(\mathbf{P}, \Omega)$ needs to be replaced by $\Gamma(\mathbf{P}, \Omega)$ in order for the self-energy to be irreducible (thus avoiding a double counting of diagrams when inserting the self-energy in the Dyson equation).

The boson and fermion self-energies then read

$$\Sigma_{\mathrm{BF}}(\mathbf{k}, \omega) = \int \frac{d\mathbf{P}}{(2\pi)^3} \int \frac{d\Omega}{2\pi} T(\mathbf{P}, \Omega) G_{\mathrm{F}}^0(\mathbf{P} - \mathbf{k}, \Omega - \omega) e^{i\Omega 0^+}, \tag{10}$$

$$\Sigma_{\mathrm{F}}(\mathbf{k}, \omega) = n_0 \Gamma(\mathbf{k}, \omega) - \int \frac{d\mathbf{P}}{(2\pi)^3} \int \frac{d\Omega}{2\pi} T(\mathbf{P}, \Omega) G_{\mathrm{B}}^0(\mathbf{P} - \mathbf{k}, \Omega - \omega) e^{i\Omega 0^+}. \tag{11}$$

Note here that we adopt a sign convention such that the $T$-matrix $T(\mathbf{P}, \Omega)$ reduces to the $T$-matrix in vacuum [87].

Since the compressibility of an ideal Bose gas in the condensed phase is infinite, the presence of a repulsive interaction between bosons is always necessary for mechanical stability. Within the Bogoliubov approximation, the normal and anomalous self-energies describing the BB interaction take the form [86]

$$\Sigma_{11} = \frac{8\pi a_{\mathrm{BB}}}{m_{\mathrm{B}}} n_0, \tag{12}$$

$$\Sigma_{12} = \frac{4\pi a_{\mathrm{BB}}}{m_{\mathrm{B}}} n_0. \tag{13}$$

The Bogoliubov approximation was used in Ref. [52] to take care of a (weak) BB interaction on top of a strong BF interaction. In this respect, we note that the Bogoliubov approximation is based on the assumption that the condensate depletion is small, a requirement that, in the absence of BF interactions, is guaranteed by the condition $n_{\mathrm{B}} a_{\mathrm{BB}}^3 \ll 1$. However, in the presence of strong BF attraction, the condensate depletion can be large even for small $n_{\mathrm{B}} a_{\mathrm{BB}}^3$.

Thus, one might envisage improving the Bogoliubov approach by adopting the Popov approximation [86], which, at $T = 0$, consists of replacing the condensate density $n_0$ with the full density $n_{\mathrm{B}}$ in Eq. (12) for the normal self-energy $\Sigma_{11}$. This is what we will do in the present work. In particular, we will see that the change from the Bogoliubov to the Popov approximation is particularly important when evaluating the stability of the mixture.

After including the BB interaction as above, the resulting expression for the normal boson self-energy has the final form

$$\Sigma_{\mathrm{B}}(\mathbf{k}, \omega) = \Sigma_{11} + \int \frac{d\mathbf{P}}{(2\pi)^3} \int \frac{d\Omega}{2\pi} T(\mathbf{P}, \Omega) G_{\mathrm{F}}^0(\mathbf{P} - \mathbf{k}, \Omega - \omega) e^{i\Omega 0^+}. \tag{14}$$

Once the bosonic and fermionic self-energies are identified, the Dyson's equations for the corresponding Green's functions read

$$G_{\mathrm{F}}^{-1}(\mathbf{k}, \omega) = G_{\mathrm{F}}^0(\mathbf{k}, \omega)^{-1} - \Sigma_{\mathrm{F}}(\mathbf{k}, \omega), \tag{15}$$

$$G_{\mathrm{B}}^{-1}(\mathbf{k}, \omega) = i\omega - \xi_{\mathbf{k}}^{\mathrm{B}} - \Sigma_{\mathrm{B}}(\mathbf{k}, \omega) + \frac{\Sigma_{12}^2}{i\omega + \xi_{\mathbf{k}}^{\mathrm{B}} + \Sigma_{\mathrm{B}}(-\mathbf{k}, -\omega)}, \tag{16}$$

where Eq. (16) results from the matrix structure of Dyson's equation in the condensed phase [88].

The number densities for fermions and bosons are then obtained from the Green's functions as [88]

$$n_{\text{F}} = \int \frac{d\mathbf{k}}{(2\pi)^3} \int_{-\infty}^{+\infty} \frac{d\omega}{2\pi} G_{\text{F}}(\mathbf{k}, \omega) e^{i\omega 0^+} \, , \tag{17}$$

$$n_{\text{B}} = -\int \frac{d\mathbf{k}}{(2\pi)^3} \int_{-\infty}^{+\infty} \frac{d\omega}{2\pi} G_{\text{B}}(\mathbf{k}, \omega) e^{i\omega 0^+} + n_0 \, . \tag{18}$$

Finally, in the condensed phase, the Hugenholtz-Pines condition [89] requires

$$\mu_{\text{B}} = \Sigma_{\text{B}} (\mathbf{k} = \mathbf{0}, \omega = 0) - \Sigma_{12} \, . \tag{19}$$

The three equations (17-19) constitute a system of non-linear integral equations in the unknowns $\mu_{\text{F}}, \mu_{\text{B}}$ and $n_0$, for given values of the densities $n_{\text{B}}, n_{\text{F}}$ and scattering lengths $a_{\text{BF}}$ and $a_{\text{BB}}$ [52]. Given the greater numerical simplicity of Eq. (19) with respect to Eqs. (17) and (18), the unknown $\mu_{\text{B}}$ is treated as dependent on $\mu_{\text{F}}$ and $n_0$ and determined by applying a bisection method to Eq. (19) for given values of $\mu_{\text{F}}$ and $n_0$. Therefore, one is effectively left with a two-dimensional system of nonlinear equations that is solved via a two-dimensional quasi-Newton method, whereby at each iteration the Jacobian is approximated according to a symmetric rank 1 algorithm [90]. Finally, we note that in the normal phase with $n_0 = 0$, which is reached for sufficiently large values of the BF coupling $g_{\text{BF}}$ [52], one has to drop the Hugenholtz-Pines condition. Equations (17) and (18) with $n_0$ set to zero are then used to determine $\mu_{\text{B}}$ and $\mu_{\text{F}}$ in the normal phase.

## 3 Mechanical stability

### 3.1 Chemical potentials and condensate fraction

Before delving into the analysis of mechanical stability, we present the results for the thermodynamic parameters $\mu_{\text{B}}, \mu_{\text{F}}, n_0$ obtained by solving the set of Eqs. (17-19). The results for these quantities have already been presented in [52]. Here, we are interested in assessing the differences introduced by replacing the Bogoliubov approximation with the Popov approximation to describe BB interactions, while always using the $T$-matrix approximation (TMA) to describe BF interactions. We focus on two concentrations with $x < 1$ (as relevant for the experiment [25]), encompassing a small concentration case ($x = 0.175$) and a large concentration case ($x = 0.9$). The specific value $x = 0.175$ is used for comparison with Refs. [48,52], where this concentration was used as a representative case of small concentrations.

Figure 3 presents the results for $\mu_{\text{B}}, \mu_{\text{F}}, n_0$ for equal masses, while Fig. 4 reports the corresponding results for the mass ratio $m_{\text{B}}/m_{\text{F}} = 0.575$ relevant to the $^{23}$Na$-^{40}$K mixture of the experiment of Ref. [25]. As discussed in [52], the increase of the BF coupling $g_{\text{BF}}$ from weak to strong values induces the formation of BF molecules at the expense of the condensate, and a transition to the normal phase ($n_0 = 0$) occurs at the critical coupling $g_{\text{BF}}^c$. The latter quantity is also identified by the singularities in the slope of the chemical potentials. Overall, one sees that the differences between the results obtained with the two approximations are generally rather small for both mass ratios. This is true in particular for the fermionic chemical potential and the condensate fraction, while, as expected, the bosonic chemical potential is the quantity that is mostly affected by the approximation that is used to describe the BB interaction.

As a benchmark for our numerical calculations, Figs. 3 and 4 also show the asymptotic expressions in both the weak and strong coupling limits of the BF interaction. In particular,

in the weak-coupling limit $g_{BF} \to -\infty$, one can use the results of Ref. [82] for the chemical potentials:

$$\mu_B = \frac{4\pi a_{BB} n_B}{m_B} + \frac{2\pi a_{BF} n_F}{m_r}\left[1 + \frac{k_F a_{BF}}{\pi}f(\delta)\right], \tag{20}$$

$$\mu_F = E_F + \frac{2\pi a_{BF} n_B}{m_r}\left[1 + \frac{4k_F a_{BF}}{3\pi}f(\delta)\right], \tag{21}$$

where $\delta = (m_B - m_F)/(m_B + m_F)$, and $f(\delta)$ is given by

$$f(\delta) = \frac{3}{4} - \frac{3}{4\delta} + \frac{3(1+\delta)(1-\delta^2)}{8\delta^2}\ln\left(\frac{1+\delta}{1-\delta}\right). \tag{22}$$

The case of equal masses is readily obtained by taking $\delta \to 0$ in Eq. (22), yielding $f(0) = 3/2$. The above equations for $\mu_B$ and $\mu_F$ can be conveniently expressed in terms of the dimensionless interactions $g_{BF}$ and $\zeta_{BB}$:

$$\frac{\mu_B}{E_F} = \frac{4x}{3\pi}\frac{m_F}{m_B}\zeta_{BB} + \frac{2}{3\pi}\frac{m_F}{m_r}\frac{1}{g_{BF}}\left[1 + \frac{1}{\pi g_{BF}}f(\delta)\right], \tag{23}$$

$$\frac{\mu_F}{E_F} = 1 + \frac{2}{3\pi}\frac{m_F}{m_r}\frac{x}{g_{BF}}\left[1 + \frac{4}{3\pi g_{BF}}f(\delta)\right]. \tag{24}$$

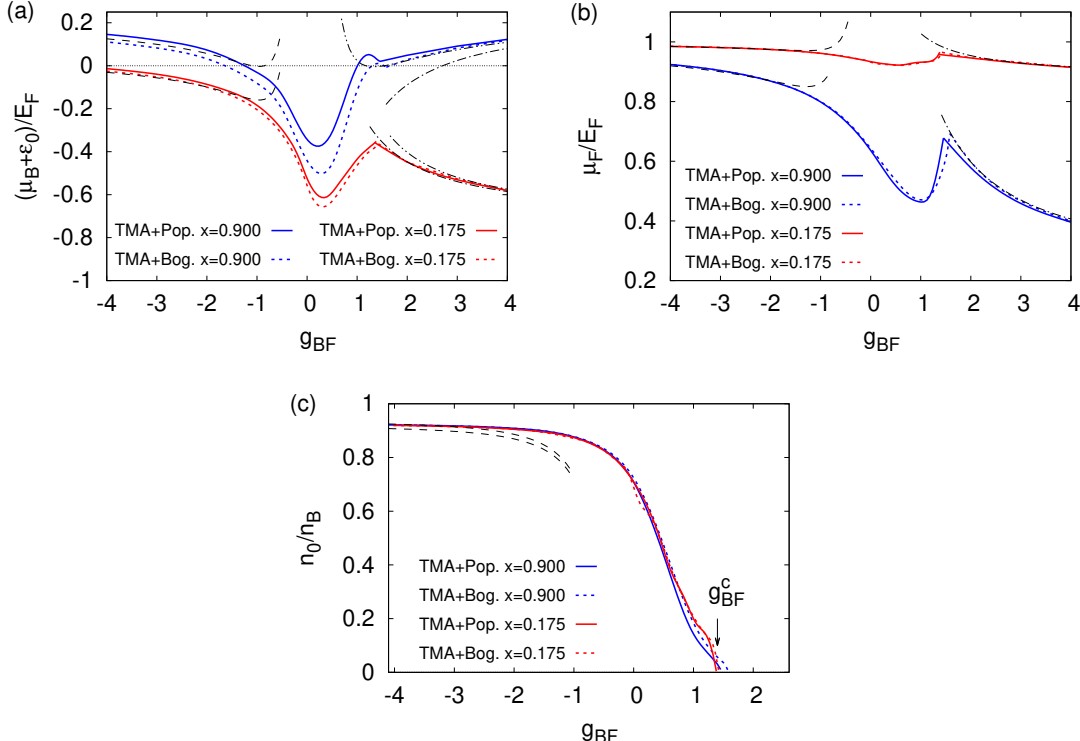

Figure 3: Bosonic (a) and fermionic (b) chemical potentials [in units of $E_F = k_F^2/(2m_F)$] and condensate fraction $n_0/n_B$ (c) versus coupling strength $g_{BF} = (k_F a_{BF})^{-1}$, computed at $n_B a_{BB}^3 = 3 \times 10^{-3}$ and mass ratio $m_B/m_F = 1$ for two representative values of $x$ within the Bogoliubov (dashed lines) and Popov (solid lines) approximations for the BB interaction. The contribution $-\epsilon_0$ due to the bound state in the two-body problem has been subtracted to $\mu_B$ for $g_{BF} > 0$. Dashed lines: weak-coupling expansions (23) for $\mu_B$, (24) for $\mu_F$, (26) for $n_0/n_B$. dash-dotted lines: strong-coupling expansions (29) for $\mu_B$, (30) for $\mu_F$; dashed double-dotted line in panel (b): next-order strong-coupling expansion (A.4) for $x = 0.9$.

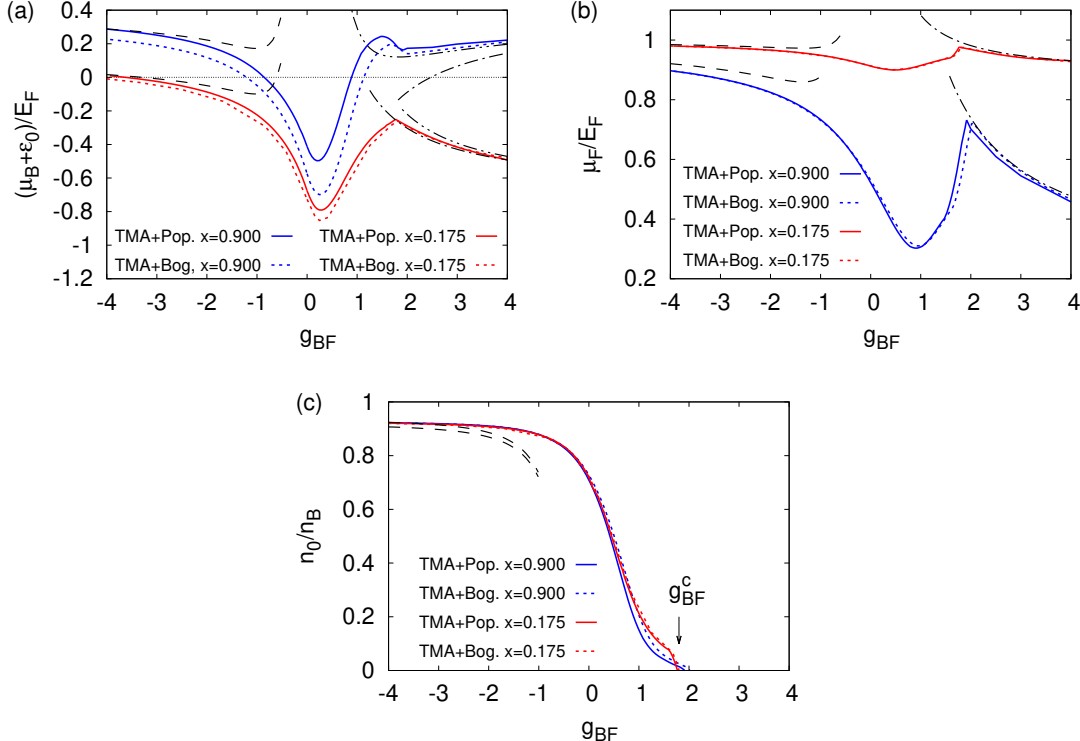

Figure 4: Bosonic (a) and fermionic (b) chemical potentials [in units of $E_F = k_F^2/(2m_F)$] and condensate fraction $n_0/n_B$ (c) versus coupling strength $g_{BF} = (k_F a_{BF})^{-1}$, computed at $n_B a_{BB}^3 = 3 \times 10^{-3}$ and mass ratio $m_B/m_F = 0.575$ for two representative values of $x$ within the Bogoliubov (dashed lines) and Popov (solid lines) approximations for the BB interaction. The contribution $-\epsilon_0$ due to the bound state in the two-body problem has been subtracted from $\mu_B$ for $g_{BF} > 0$. Dashed lines: weak-coupling expansions (23) for $\mu_B$, (24) for $\mu_F$, (25) for $n_0/n_B$. dash-dotted lines: strong-coupling expansions (29) for $\mu_B$, (30) for $\mu_F$; dashed double-dotted line in panel (a): next-order strong-coupling expansion (A.4) for $x = 0.9$.

Regarding the condensate fraction, Ref. [71] reports an expression involving a double integral that is parametrically dependent on the mass ratio $m_B/m_F$ (see Eqs. (39) and (40) of Ref. [71]). We have found that this double integral can be calculated in a closed form, yielding

$$\frac{n_0}{n_B} = 1 - \frac{8}{3}\sqrt{\frac{n_B a_{BB}^3}{\pi}} - \frac{1}{(\pi g_{BF})^2}\frac{m_B/m_F + 1}{m_B/m_F - 1}\ln\left(\frac{m_B}{m_F}\right), \tag{25}$$

which, for equal masses, reduces to

$$\frac{n_0}{n_B} = 1 - \frac{8}{3}\sqrt{\frac{n_B a_{BB}^3}{\pi}} - \frac{2}{(\pi g_{BF})^2}. \tag{26}$$

In Figs. 3 and 4 one sees that the weak-coupling expressions (23−25) provide a good approximation already at $g_{BF} \simeq -2.5$.

In the strong-coupling limit $g_{BF} \to +\infty$, we can use the results of Ref. [51] to leading

order in $a_{\mathrm{BF}}$

$$\mu_{\mathrm{B}} = -\epsilon_0 + \frac{\left(6\pi^2 n_{\mathrm{B}}\right)^{2/3}}{2M} - \frac{\left[6\pi^2\left(n_{\mathrm{F}} - n_{\mathrm{B}}\right)\right]^{2/3}}{2m_{\mathrm{F}}} - \frac{4\pi a_{\mathrm{BF}}}{m_{\mathrm{r}}} n_{\mathrm{B}} + \frac{4\pi a_{\mathrm{BF}}}{m_{\mathrm{r}}}\left(n_{\mathrm{F}} - n_{\mathrm{B}}\right), \qquad (27)$$

$$\mu_{\mathrm{F}} = \frac{\left[6\pi^2\left(n_{\mathrm{F}} - n_{\mathrm{B}}\right)\right]^{2/3}}{2m_{\mathrm{F}}} + \frac{4\pi a_{\mathrm{BF}}}{m_{\mathrm{r}}} n_{\mathrm{B}}, \qquad (28)$$

corresponding to the dimensionless expressions

$$\frac{\mu_{\mathrm{B}} + \epsilon_0}{E_{\mathrm{F}}} = \frac{m_{\mathrm{F}}}{M} x^{2/3} - (1-x)^{2/3} + \frac{4}{3\pi} \frac{m_{\mathrm{F}}}{m_{\mathrm{r}}} \frac{1-2x}{g_{\mathrm{BF}}}, \qquad (29)$$

$$\frac{\mu_{\mathrm{F}}}{E_{\mathrm{F}}} = (1-x)^{2/3} + \frac{4}{3\pi} \frac{m_{\mathrm{F}}}{m_{\mathrm{r}}} \frac{x}{g_{\mathrm{BF}}}. \qquad (30)$$

Note that in the strong-coupling limit $\mu_{\mathrm{B}}$ becomes independent of the BB interaction. This is trivial within the Bogoliubov approximation since the Bogoliubov self-energies (12) and (13) vanish identically when $n_0 = 0$ ($n_0$ vanishes at a critical coupling $g_c$ and can thus be set to zero in the strong-coupling approximations). Within the Popov approximation, the self-energy $\Sigma_{11}$ does not vanish (it remains constant), but it is negligible with respect to $\Sigma_{\mathrm{BF}}$, which grows like $\sqrt{\epsilon_0}$ [51].

The numerical results for $\mu_{\mathrm{B}}$ and $\mu_{\mathrm{F}}$ in Figs. 3 and 4 (panels (a) and (b)) show very good agreement with the corresponding strong-coupling expressions (27−28), except for $\mu_{\mathrm{B}}$ in the case $x = 0.9$. To explain this discrepancy, we have pushed the strong-coupling expansion of [51] to second order in the expansion parameter $k_{\mathrm{F}} a_{\mathrm{BF}}$ (see A for details). It is displayed as a dash-double-dotted line in Figs. 3(b) and 4(b), restoring very good agreement between the numerical and analytical results also for $x = 0.9$. For low concentrations, $x = 0.175$, the second-order correction instead gives a smaller contribution, which apparently slightly worsens the good agreement reached by the first-order expansion already at $g_{\mathrm{BF}} \simeq 1.5$. However, on closer examination, one notices that the inclusion of the second-order term makes the approaching of the strong-coupling benchmark by the numerical data monotonous, as it should. In contrast, the good agreement of $\mu_{\mathrm{F}}$ with its strong-coupling benchmark already at first order in $k_{\mathrm{F}} a_{\mathrm{BF}}$ is explained by the absence of second-order terms in its perturbative expansion (28).

Returning to the first-order strong-coupling expansions, it is useful to recast Eq. (27) and (28) in a form that highlights the transmutation of the original BF mixture into a Fermi-Fermi (FF) mixture of fermionic BF dimers (i.e., molecules made by binding a boson with a fermion) and unpaired fermions (recalling that $n_{\mathrm{B}} \leq n_{\mathrm{F}}$ in the present work). We describe this transmutation as a "fermionization" of the original BF mixture. Equations (27) and (28) can indeed be rewritten as

$$\mu_{\mathrm{B}} = -\epsilon_0 + \frac{\left(6\pi^2 n_{\mathrm{D}}\right)^{2/3}}{2M} + \frac{2\pi a_{\mathrm{DF}}}{m_{\mathrm{DF}}} n_{\mathrm{UF}} - \left[\frac{\left(6\pi^2 n_{\mathrm{UF}}\right)^{2/3}}{2m_{\mathrm{F}}} + \frac{2\pi a_{\mathrm{DF}}}{m_{\mathrm{DF}}} n_{\mathrm{D}}\right], \qquad (31)$$

$$\mu_{\mathrm{F}} = \frac{\left[6\pi^2 n_{\mathrm{UF}}\right]^{2/3}}{2m_{\mathrm{F}}} + \frac{2\pi a_{\mathrm{DF}}}{m_{\mathrm{DF}}} n_{\mathrm{D}}, \qquad (32)$$

where we have introduced the dimer density $n_{\mathrm{D}} = n_{\mathrm{B}}$, the unpaired fermion density $n_{\mathrm{UF}} = n_{\mathrm{F}} - n_{\mathrm{B}}$, the dimer-fermion (DF) reduced mass $m_{\mathrm{DF}} = M m_{\mathrm{F}}/(M + m_{\mathrm{F}})$, and the DF scattering length [46]

$$a_{\mathrm{DF}} = \frac{(1 + m_{\mathrm{F}}/m_{\mathrm{B}})^2}{1/2 + m_{\mathrm{F}}/m_{\mathrm{B}}} a_{\mathrm{BF}}. \qquad (33)$$

The interpretation of Eqs. (31) and (32) is fairly intuitive. When a Fermi atom is added to the FF mixture of BF dimers and unpaired fermions, it will be placed at the Fermi surface of the unpaired fermions, with kinetic energy given by the first term in Eq. (32) and a mean-field repulsion with dimers given by the second term in Eq. (32). When a boson is added to the mixture, a new dimer will form, with an energy gain due to binding $-\epsilon_0$, kinetic energy $\left(6\pi^2 n_{\mathrm{D}}\right)^{2/3}/2M$, and mean-field repulsion $\frac{2\pi a_{\mathrm{DF}}}{m_{\mathrm{DF}}}n_{\mathrm{UF}}$, corresponding to the first three terms in Eq. (31). The formation of the dimer will simultaneously require the removal of an unpaired fermion from the Fermi surface, corresponding to the last two terms in Eq. (31).

We finally remark that, for equal masses ($m_{\mathrm{B}} = m_{\mathrm{F}}$), Eq. (33) yields $a_{\mathrm{DF}} = 8/3a_{\mathrm{BF}}$ [91], to be compared with the exact value $a_{\mathrm{DF}} = 1.18a_{\mathrm{BF}}$ [92, 93]. It is indeed known from the analysis of Ref. [94] of the three-body problem that the correct dimer-fermion scattering length is recovered by summing all diagrams reconstructing the exact $T$-matrix for the dimer-fermion scattering (the so-called $T_3$). The present non-self-consistent $T$-matrix approximation for the boson-fermion self-energy includes only the first diagram of the infinite series of diagrams that build up the exact $T_3$ (see also appendix B of [62]), in the very same way as the Born approximation for the exact $T$-matrix in the quantum theory of scattering by a potential [85]. For this reason, expression (33) is only a Born approximation to the exact result for $a_{\mathrm{DF}}$. Implementing the full $T_3$ in a many-body setting in a computationally manageable way is still an open problem (proposals for implementing the $T_3$ in a many-body theory have been put forward in [91] and [95] for the related problem of polarized Fermi gases in the superfluid phase, but only at a formal level).

## 3.2 Stability matrix

The mechanical stability of a gas or liquid mixture is assessed by looking at the (Hessian) matrix $\mathcal{M}$ of the second order derivatives of the free energy with respect to the densities of the two species, which can be cast in the following form [57, 75]

$$\mathcal{M} = \begin{pmatrix} \dfrac{\partial \mu_{\mathrm{F}}}{\partial n_{\mathrm{F}}} & \dfrac{\partial \mu_{\mathrm{F}}}{\partial n_{\mathrm{B}}} \\[2mm] \dfrac{\partial \mu_{\mathrm{B}}}{\partial n_{\mathrm{F}}} & \dfrac{\partial \mu_{\mathrm{B}}}{\partial n_{\mathrm{B}}} \end{pmatrix}, \qquad \left[ \mathcal{M}_{\alpha\beta} = \frac{\partial \mu_\alpha}{\partial n_\beta}, \quad \alpha, \beta = \mathrm{F}, \mathrm{B} \right]. \tag{34}$$

Mechanical stability requires $\mathcal{M}$ to be positive definite, which is equivalent to the following conditions [43, 69, 87]:

$$\frac{\partial \mu_{\mathrm{F}}}{\partial n_{\mathrm{F}}} + \frac{\partial \mu_{\mathrm{B}}}{\partial n_{\mathrm{B}}} > 0, \quad \text{and} \quad \frac{\partial \mu_{\mathrm{F}}}{\partial n_{\mathrm{F}}} \frac{\partial \mu_{\mathrm{B}}}{\partial n_{\mathrm{B}}} - \frac{\partial \mu_{\mathrm{F}}}{\partial n_{\mathrm{B}}} \frac{\partial \mu_{\mathrm{B}}}{\partial n_{\mathrm{F}}} > 0. \tag{35}$$

We note that $(\partial \mu_{\mathrm{F}}/\partial n_{\mathrm{B}})(\partial \mu_{\mathrm{B}}/\partial n_{\mathrm{F}}) > 0$, due to the thermodynamic identity

$$(\partial \mu_{\mathrm{F}}/\partial n_{\mathrm{B}}) = (\partial \mu_{\mathrm{B}}/\partial n_{\mathrm{F}}).$$

The fulfillment of the second inequality in (35) then implies that $(\partial \mu_{\mathrm{F}}/\partial n_{\mathrm{F}})$ and $(\partial \mu_{\mathrm{B}}/\partial n_{\mathrm{B}})$ must have the same sign. The first inequality in (35) can then be replaced by $(\partial \mu_{\mathrm{F}}/\partial n_{\mathrm{F}}) > 0$, a requirement that, due to Fermi pressure, is always satisfied in our mixture. Therefore, it is only the second inequality in (35) that in practice rules the stability of the mixture.

In order to compute the stability matrix (34) and the related stability conditions (35), a straightforward finite difference evaluation of the first-order derivatives appearing in (34) is performed: the input densities $n_{\mathrm{B}}$ and $n_{\mathrm{F}}$ are slightly changed one at a time and the shifted values of $\mu_{\mathrm{B}}$ and $\mu_{\mathrm{F}}$ are obtained by solving Eqs. (17-19) accordingly.

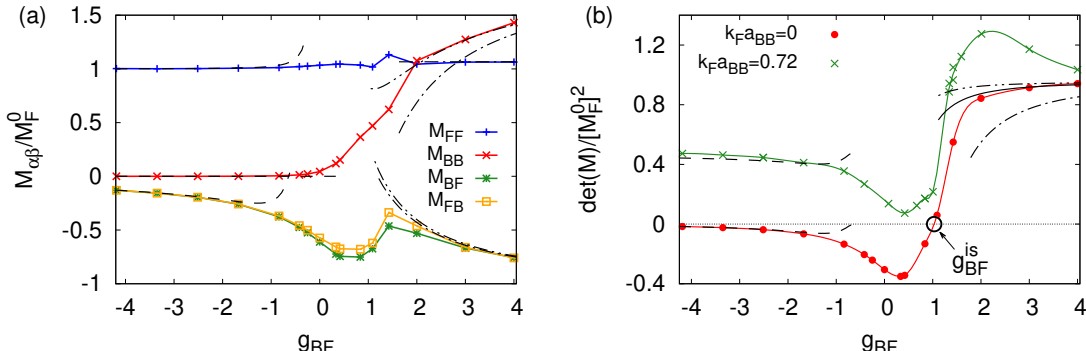

Figure 5: (a) Coefficients $\mathcal{M}_{\alpha\beta}$ of the stability matrix as a function of coupling $g_{BF} = (k_F a_{BF})^{-1}$ in the case of equal masses, boson concentration $x \equiv n_B/n_F = 0.175$, and vanishing BB repulsion. The non-interacting Fermi gas value $\mathcal{M}_F^0 = \frac{2\pi^2}{m_F k_F}$ is used for normalization purposes. (b) Determinant of the stability matrix (in units of $[\mathcal{M}_F^0]^2$) as a function of coupling for equal masses and vanishing BB repulsion (red circles with interpolating cubic spline) as well as finite BB repulsion $\zeta_{BB} = k_F a_{BB} = 0.72$, corresponding to $n_B a_{BB}^3 = 1.1 \times 10^{-3}$ (green crosses with interpolating cubic spline). Dashed lines: weak-coupling benchmarks (36); dash-dotted lines: strong-coupling benchmarks (39); dash-double-dotted lines: next-order strong-coupling benchmarks resulting from Eq. (A.5). Full black line in panel (b): strong-coupling expansion for $\det\mathcal{M}_{\text{strong}}$ using the exact value $a_{DF} = 1.18 a_{BF}$ in Eq.(31). Empty circle: coupling $g_{BF}^{\text{is}}$ at which the mixture becomes intrinsically stable.

The weak and strong coupling limits of the matrix elements $\mathcal{M}_{\alpha\beta}$ provide a check on the numerical calculations, as well as physical insight into the problem. For weak coupling strength, differentiation of Eqs. (20) and (21) with respect to $n_F$ or $n_B$ yields

$$\mathcal{M}_{\text{weak}} = \frac{2\pi^2}{m_F k_F} \begin{pmatrix} 1 + \frac{4x}{9\pi^2} \frac{m_F}{m_r} \frac{f(\delta)}{g_{BF}^2} & \frac{1}{\pi} \frac{m_F}{m_r} \frac{1}{g_{BF}} \left[ 1 + \frac{4}{3\pi} \frac{f(\delta)}{g_{BF}} \right] \\ \frac{1}{\pi} \frac{m_F}{m_r} \frac{1}{g_{BF}} \left[ 1 + \frac{4}{3\pi} \frac{f(\delta)}{g_{BF}} \right] & \frac{2}{\pi} \frac{m_F}{m_B} \zeta_{BB} \end{pmatrix}. \tag{36}$$

We see from the stability matrix (36) that, in the weak-coupling limit, a finite BB repulsion is required for the matrix to be positive definite. In particular, positivity of the determinant requires

$$\zeta_{BB} > \frac{1}{2\pi} \frac{m_B m_F}{m_r^2} \frac{1}{g_{BF}^2} \frac{\left[ 1 + \frac{4}{3\pi} \frac{f(\delta)}{g_{BF}} \right]^2}{1 + \frac{4x}{9\pi^2} \frac{m_F}{m_r} \frac{f(\delta)}{g_{BF}^2}}, \tag{37}$$

which, neglecting terms of order higher than $1/g_{BF}^2$, reduces to the mean-field stability criterion [69, 72]

$$\zeta_{BB} > \frac{1}{2\pi} \frac{m_B m_F}{m_r^2} \frac{1}{g_{BF}^2} . \tag{38}$$

This mean-field condition can be interpreted as the requirement that, in the long-wavelength limit [96], the direct BB repulsion overcomes the effective attraction between bosons induced by the fermionic component, so that the overall effective interaction between bosons is repulsive [60, 87].

Note that the beyond-mean-field corrections included in (37) act to reduce the minimum value of the boson repulsion required for stability in the weak-coupling regime.

Differentiation of Eqs. (27) and (28) yields instead the strong-coupling approximation

$$\mathcal{M}^{\text{strong}} = \frac{2\pi^2}{m_F k_F} \begin{pmatrix} \frac{1}{(1-x)^{1/3}} & -\frac{1}{(1-x)^{1/3}} + \frac{2}{\pi}\frac{m_F}{m_r}\frac{1}{g_{BF}} \\ -\frac{1}{(1-x)^{1/3}} + \frac{2}{\pi}\frac{m_F}{m_r}\frac{1}{g_{BF}} & \frac{1}{(1-x)^{1/3}} + \frac{m_F}{M x^{1/3}} - \frac{4}{\pi}\frac{m_F}{m_r}\frac{1}{g_{BF}} \end{pmatrix}, \tag{39}$$

which implies

$$\lim_{g_{BF}\to+\infty} \det \mathcal{M}^{\text{strong}} = \left(\frac{2\pi^2}{m_F k_F}\right)^2 \frac{1}{(1-x)^{1/3}}\frac{m_F}{M x^{1/3}} > 0, \tag{40}$$

that is, the stability of the mixture independently of the BB repulsion in the strong-coupling limit of the BF interaction. Physically, this is expected since in this limit the original Bose-Fermi mixture "fermionizes" in a two-component FF mixture of molecules and unpaired fermions.

Figure 5 reports an example of the results obtained by a numerical evaluation of the stability matrix. Specifically, Fig. 5(a) shows the matrix elements $\mathcal{M}_{\alpha\beta}$ versus coupling for concentration $x = 0.175$, equal masses, and vanishing BB repulsion ($\zeta_{BB} = 0$), while panel (b) reports the corresponding determinant for $\zeta_{BB} = 0$ (red circles) as well as for a finite repulsion $\zeta_{BB} = 0.72$ (green crosses) taken into account within Popov theory.

We notice that the diagonal coefficients $\mathcal{M}_{FF}$ and $\mathcal{M}_{BB}$ remain approximately constant from the weak-coupling limit to $g_{BF} = 0$. In this region, the mixture is unstable in the absence of BB repulsion, as can be seen from the negative determinant in panel (b) for the case $\zeta_{BB} = 0$ (red circles). Beyond the unitary limit $g_{BF} = 0$, the coefficient $\mathcal{M}_{BB}$ starts to increase significantly, leading to a rapid change in the slope of the $\det\mathcal{M}$ vs coupling, and eventually to a stable mixture when $g_{BF} \gtrsim 1$, as signaled by $\det\mathcal{M} > 0$. This rapid change can be associated with a correspondingly rapid decrease in the condensate fraction past unitarity, as one can see in Fig. 3 of Ref. [52] and in our Figs. 3 and 4, which can, in turn, be ascribed to a rapid increase in the formation of BF dimers (molecules) that subtract bosons from the condensate [52].

This fermionization process makes the mixture intrinsically stable (that is, without the need for a BB repulsion), and its onset is identified by the coupling $g_{BF}^{\text{is}}$ above which $\det\mathcal{M} > 0$. For the case $\zeta_{BB} = 0$ (red circles) in Fig. 5(b), we find $g_{BF}^{\text{is}} = 1.03$. Interestingly, such stabilization of the mixture occurs (slightly) before the critical coupling $g_{BF}^c$ (=1.42, as shown in Fig. 3(c)) at which the condensate fraction vanishes and above which the original BF mixture is effectively replaced by a composite FF mixture of BF molecules and unpaired fermions (see sec. 3.1) [48, 51, 52].

For generic BF coupling values, a sufficiently strong BB repulsion is instead needed to stabilize the mixture. For example, one sees in Fig. 5(b) that for a BF mixture with equal masses and concentration $x = 0.175$ a BB repulsion $\zeta_{BB} \gtrsim 0.7$ guarantees the stability for all BF couplings (green crosses).

It is also interesting to note that including the next-order correction in the strong-coupling expansion for $\mu_B$ is necessary to recover in Fig. 5 a good agreement between the strong-coupling benchmarks and the numerical data for $\mathcal{M}_{BB}$ and $\det\mathcal{M}$, confirming the importance of this correction even for the case of small concentrations ($x = 0.175$) considered in Fig. 5. Furthermore, in Fig. 5(b) one sees that our numerical results for $\det\mathcal{M}$ nicely connect at $g_{BF} \simeq 2$ with the curve for $\det\mathcal{M}$ obtained by the exact theory in the strong-coupling limit, namely, the strong-coupling expansion for $\det\mathcal{M}_{\text{strong}}$ with the exact value $a_{DF} = 1.18a_{BF}$ inserted in Eq.(31). We speculate that this is due to a compensation in this coupling region between the diagrams necessary to recover the correct dimer-fermion scattering length [94] in the strong-coupling limit and other higher-order diagrams not considered in the present work (such as, e.g., self-consistency corrections in the propagators).

Finally, we notice that the off-diagonal matrix elements respect the relation $\mathcal{M}_{FB} = \mathcal{M}_{BF}$, which should be obeyed by an exact theory, except for small violations at intermediate couplings. This provides a consistency check for our theoretical approach.

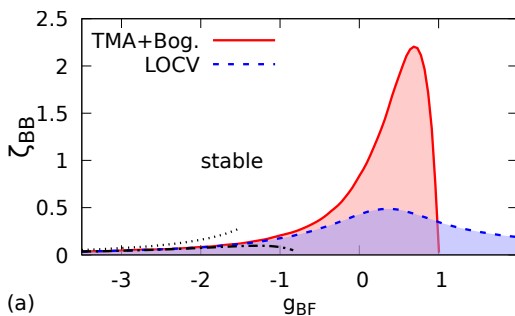
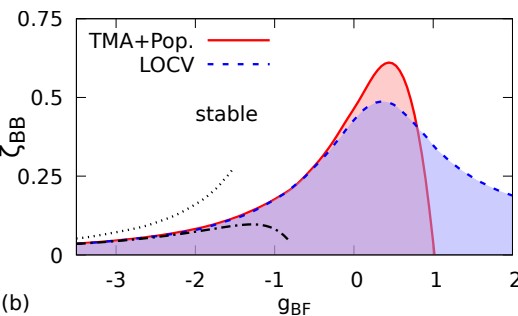

Figure 6: Stability phase diagram in the plane $\zeta_{BB}(\equiv k_F a_{BB})$ vs. $g_{BF}(\equiv (k_F a_{BF})^{-1})$ for equal masses at concentration $x = 0.175$ within (a) the $T$-matrix plus Bogoliubov approximation (TMA+Bog.) and (b) the $T$-matrix plus Popov approximation (TMA+Pop.). The results obtained with the lowest-order constrained variational approximation [43] (LOCV), the mean-field approximation (38) (dotted line), and the weak-coupling expansion (37) (dash-dotted line) are also reported for comparison. Shaded areas: unstable regions according to TMA+Bogoliubov/Popov (red) or LOCV (blue) approximations.

## 3.3 Stability phase diagram

We now discuss the stability phase diagrams in the plane $\zeta_{BB}$ vs. $g_{BF}$ for given concentrations and mass ratios. To this end, for a given BF coupling, we gradually increase the BB repulsion $\zeta_{BB}$ until the stability conditions (35) are met (as already mentioned, it is actually the second condition in (35) to rule in practice the stability).

We first focus on the specific case $x = 0.175$ and $m_B = m_F$ to compare our results with those obtained with other approximations. Specifically, in Fig. 6 we compare the results obtained with the $T$-matrix + Bogoliubov approximation and $T$-matrix + Popov approximation (full lines in panels (a) and (b), respectively) with those obtained by the lowest-order constrained variational approximation (LOCV) [43] (dashed lines), as well as with mean-field (38) (dotted lines) and weak-coupling (37) (dash-dotted line) approximations.

One first notices that both the TMA + Bogoliubov and TMA + Popov approximations correctly predict an intrinsically stable mixture above the coupling $g_{BF}^{is} \simeq 1$, as expected due to the fermionization of the BF mixture discussed above. In contrast, the LOCV approach requires a finite BB repulsion even in the molecular limit.

This is because, as also pointed out by the authors of Ref. [43], the LOCV approximation is no longer applicable in the strong-coupling limit, since it is based on a wavefunction that takes into account BF pairing only in Jastrow terms multiplying a Slater determinant of unpaired fermions. As a consequence, the nodal surface of the variational wave function becomes inadequate when molecules form. Interestingly, the LOCV approximation was estimated in Ref. [43] to break down at $g_{BF} \simeq 1$, which corresponds to the coupling strength where the BF mixture becomes intrinsically stable according to our calculations.

One also notices that, in the region around unitarity, the TMA + Bogoliubov approximation predicts values significantly higher than those obtained with both the TMA + Popov and LOCV approximations for the critical BB repulsion $\zeta_{BB}^c$ below which the mixture becomes unstable. We have already mentioned in Sec. 2.2 that the Bogoliubov approximation underestimates the BB repulsion when the condensate depletion becomes significant. In Fig. 6 one sees that the effect on the stability is quite dramatic.

The phase diagram obtained with the TMA + Bogoliubov approximation also disagrees with the Quantum Monte Carlo simulations of Ref. [48] in the region around unitarity: in that

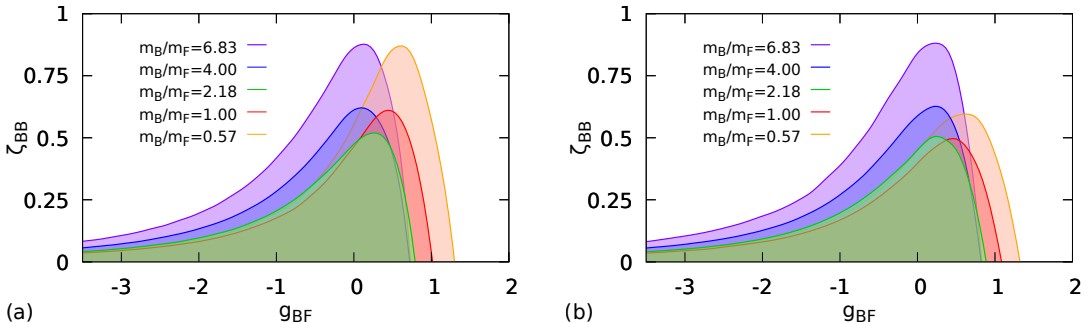

Figure 7: Stability phase diagram obtained with the $T$-matrix + Popov approximation for several values of the mass ratios at concentrations (a) $x = 0.175$ and (b) $x = 0.9$. The different shaded areas correspond to the unstable regions for each mass ratio, which can be identified from the legend.

work, a BB repulsion $\zeta_{BB} = 1$ was used in all simulations and no evidence of clustering or collapse was found in the region around unitarity where the TMA + Bogoliubov approximation instead predicts $\zeta_{BB}^c > 1$. In the rest of the paper we will thus adopt the TMA + Popov approximation to address stability.

We now turn to the discussion of the stability phase diagram for different mass ratios $m_B/m_F$ and two representative concentrations $x < 1$. We include the experimentally relevant cases of $^{23}$Na $- {}^{40}$K, $^{87}$Rb $- {}^{40}$K, and $^{41}$K $- {}^6$Li mixtures, which correspond to the mass ratios $m_B/m_F = 0.57$, 2.18, and 6.83, respectively. No significant qualitative changes are observed in Fig. 7 compared to the case of equal masses.

Quantitatively, we observe that, among the different cases considered in Fig. 7, the mixture $^{87}$Rb $- {}^{40}$K ($m_B/m_F = 2.18$) is the overall optimal one for stability (i.e., it generally requires lower BB repulsions across the whole range of BF interactions that span the evolution of the BF mixture from weak to strong coupling). Approximately, this can be understood on the weak-coupling side ($g_{BF} \lesssim -1$) by considering the mean-field stability condition (38), which, as a function of $m_B/m_F$, reaches its minimum when $m_B/m_F = 1$ and is very flat between 0.5 and 2, thus indicating mixtures with $m_B/m_F$ slightly below 2 as the most stable. In contrast, on the strong-coupling side ($g_{BF} \gtrsim 1$), the mixture becomes intrinsically stable beyond the coupling $g_{BF}^{is}$ at which the critical BB repulsion vanishes.

The quantity $g_{BF}^{is}$ is reported in Fig. 8(a) as a function of the mass ratio $m_B/m_F$, where it shows a very broad minimum between 2 and 10, thus signaling mixtures with mass ratios slightly above 2 as the most stable. As a consequence, taking into account both the weak and strong coupling behaviors of $\zeta_{BB}^c$, mixtures with $m_B/m_F \sim 2$ are a good compromise for stability.

Focusing on the $^{23}$Na $- {}^{40}$K mixture studied in the experimental work [25], one sees that BB repulsions $\zeta_{BB}$ as high as $0.6 - 0.8$ (depending on the concentration of bosons $x$) are required to guarantee stability for all $g_{BF}$. Using the values $a_{BB} = 28$Å and $k_F = 9.4\,\mu\mathrm{m}^{-1}$ reported in [25], we obtain a value of $\zeta_{BB} = 2.6 \times 10^{-2}$, approximately 20 times smaller than the minimum value of $\zeta_{BB}$ required for stability across the whole phase diagram. Considering that mechanical collapse was not observed during the experiment [25], we conclude that the timescale of the experiment was short enough to avoid mechanical collapse, as also argued in [25].

Returning to Fig. 8(a), it is interesting to make a direct comparison between the coupling of intrinsic stability $g_{BF}^{is}$ and the critical coupling $g_{BF}^c$ of the transition to the normal phase [46] as a function of the mass ratio $m_B/m_F$. Fig. 8(a) reveals that $g_{BF}^{is} < g_{BF}^c$ even for generic mass ratios, extending what we previously found for equal masses. The regime of intrinsic stability

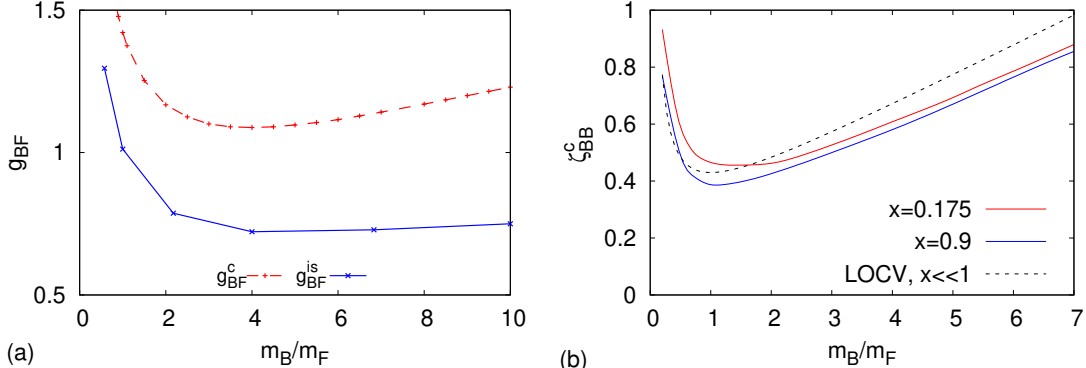

Figure 8: (a): Coupling $g_{BF}^{is}$ above which the BF mixture is intrinsically stable (full line) and critical coupling $g_{BF}^{c}$ from Ref. [46] for the transition to the normal phase (dashed line) as a function of the mass ratio $m_B/m_F$ at concentrations $x = 0.175$ and $x = 0.15$, respectively. (b): Minimum value of the BB repulsion $\zeta_{BB}^{c}$ required for stability at unitarity ($g_{BF} = 0$) versus mass ratio $m_B/m_F$ for two representative values of $x$. The outcome of the LOCV calculation [43] for small $x$ is also shown.

thus starts already in the condensed phase, somewhat anticipating the fully fermionized phase that occurs past the critical coupling $g_{BF}^{c}$.[1] For all explored mass ratios, we have verified that the fermionized phase with $n_0 = 0$ remains stable for all coupling strengths $g > g_{BF}^{c}$. In this respect, it is interesting to note that, for a point-like FF mixture, perturbative [99] and Quantum Monte Carlo calculations [100] predict a phase-separation instability at sufficiently large FF repulsions. Our results indicate that, for the effective FF mixture originating from a BF mixture at large BF attractions, such instability is preempted by the transition to the condensed phase.

Finally, in Fig. 8(b) we focus on unitarity and analyze the critical BB repulsion $\zeta_{BB}^{c}$ for stability as a function of the mass ratio for two representative concentration values. We see that in this case the mixtures with $1 \lesssim m_B/m_F \lesssim 2$ are the most stable.

# 4 Conclusions and outlook

In this work we have analyzed at zero temperature the mechanical stability of resonant Bose-Fermi mixtures with $n_B < n_F$, employing a many-body diagrammatic approach that was validated by fixed-node Quantum Monte Carlo calculations [52] and is also supported by recent experimental observations [25]. In particular, the BF interaction was treated within the same $T$-matrix formalism used in [52]. For the BB repulsion, we have found that upgrading from the Bogoliubov approximation to the Popov approximation is crucial to obtain stability results consistent with the outcomes of fixed node Quantum Monte Carlo simulations [48].

The stability phase diagrams we have obtained indicate that mixtures with boson-to-fermion mass ratios near two, such as the $^{87}Rb-^{40}K$ mixture, exhibit optimal stability over the full range of BF couplings. In contrast, applying our results to the $^{23}Na-^{40}K$ mixture used in a recent experiment [25] reveals that the BB repulsion was more than an order of magnitude

---

[1]Here we identify the fully fermionized phase with the absence of a bosonic condensate, assuming in practice that the depletion of the condensate is solely determined by the formation of molecules in the strong coupling region $g_{BF} > g_{BF}^{c}$. In this way, we circumvent the problem of the blurred definition of the number of molecules (from which the number of unpaired non-condensed bosons would be obtained by difference from $n_B$) away from the strong-coupling limit $g_{BF} \to +\infty$. For possible definitions (prompted by specific experimental protocols to measure the number of molecules) in the analogous problem emerging in the BCS-BEC crossover, see [97] and [98].

lower than the threshold required for stability. This discrepancy suggests that the absence of collapse in the experiment may be attributed to the short experimental timescales, which did not allow the mechanical instability to fully develop.

Our calculations show that even for the most favorable mass ratios $m_B/m_F \simeq 2$ a BB repulsion as large as $\zeta_{BB} = k_F a_{BB} \simeq 0.5$ is necessary to have a stable BF mixture for all BF couplings. Such large values of $\zeta_{BB}$ would require quite a favorable configuration in which the BF Feshbach resonance takes place at magnetic fields corresponding to the positive side of a BB resonance occurring close to the BF resonance.

On the other hand, our finding that the intrinsic stability coupling $g_{BF}^{is}$ precedes the critical coupling $g_{BF}^c$ indicates that the quantum phase transition separating the condensed and molecular phases is within the stable region of the mixture even in the absence of any BB repulsion. Thus, one can envisage the following experimental protocol: i) magneto-associate BF Feshbach molecules by sweeping through the resonance with a magnetic field ramp as e.g. in [6,25]; ii) remove the remaining unpaired bosonic atoms (for example, by a radio-frequency excitation to a state which is not trapped); iii) sweep back adiabatically the magnetic field until the critical coupling $g_{BF}^c$ is crossed and the formation of a bosonic condensate is observed.

In this way, the predicted quantum phase transition and the onset of a bosonic condensate out of the effective FF mixture corresponding to the molecular phase could be experimentally observed in a mechanically stable configuration, without the need for particularly favorable BF and BB Feshbach resonance configurations. Looking ahead, it would be worthwhile to extend the present zero-temperature analysis to finite temperatures, where thermal fluctuations could further influence the competition between condensation and pairing, as well as the stability phase diagram. Second, a fully dynamical study, possibly through time-dependent simulations, could elucidate the transient behavior leading to collapse, thus providing deeper insight into the experimental observations of Ref. [25]. In a further line of investigation, by incorporating higher-order interaction effects beyond the current diagrammatic framework, one could confirm (or disprove) the purported compensation between the diagrams necessary to recover the correct dimer-fermion scattering length [94] and self-consistency corrections in the propagators.

Finally, we wish to comment on the case $n_B > n_F$, which we did not address in the present work. The present theoretical approach could be used in principle to investigate also this case. However, we believe that the inclusion of the diagrams necessary to recover the correct dimer-atom scattering length would be crucial in this case, in particular when approaching the strong-coupling limit. This is because for $n_B > n_F$ the excess atoms are bosons. In this case, the solution of the three-body problem of a BF-dimer which scatters with a bosonic atom [101] shows that the dimer-boson scattering length $a_{DB}$ strongly depends on the three-body parameter $a^*$ corresponding to the value of $a_{BF}$ at which the energy of a trimer made of two bosons and one fermion is equal to the atom-dimer scattering threshold energy. In particular, $a_{DB}$ can be positive or negative depending on the ratio $a_{BF}/a^*$ (and can even display resonances). In addition, also the Bose-Bose repulsion enters in a non-trivial way in the three-body problem by shifting the position of the resonances of $a_{DB}$ [102]. Clearly these three-body features of the dimer-boson scattering length, which will be crucial in determining the mechanical stability of the mixture when approaching the molecular limit, will be out of reach of the application of the present $T$-matrix approximation to the case $n_B > n_F$. This is because the dimer-boson scattering would be treated only at the level of the Born approximation, with no dependence at all on the three-body parameter $a^*$.

All the numerical data necessary to reproduce Figs. 3-8 are available online [103].

## Acknowledgments

We acknowledge the use of the parallel computing cluster of the Open Physics Hub at the Department of Physics and Astronomy of the University of Bologna.

**Funding information**   L.P. and P.P. acknowledge financial support from the Italian Ministry of University and Research (MUR) under project PRIN2022, Contract No. 2022523NA7. P.P. also acknowledges financial support from the European Union - Next Generation EU through MUR projects PE0000023-NQSTI (Italy) and PNRR - M4C2 - I1.4 Contract No. CN00000013.

## A   Second-order contribution to the strong-coupling expansion of $\mu_{\mathrm{B}}$

The strong coupling expressions (27) and (28) contain the Hartree self-energies for BF dimers and unpaired atoms, provided by the analysis in [51]

$$\Sigma_{\mathrm{CF}} = \frac{4\pi a_{\mathrm{BF}}}{m_{\mathrm{r}}} n^0_{\mu_{\mathrm{F}}}, \qquad \Sigma^0_{\mathrm{F}} = \frac{4\pi a_{\mathrm{BF}}}{m_{\mathrm{r}}} n_{\mathrm{B}}, \tag{A.1}$$

$n^0_{\mu_{\mathrm{F}}}$ being the density of a non-interacting Fermi gas with chemical potential $\mu_{\mathrm{F}}$. At the lowest order in $a_{\mathrm{BF}}$, $n^0_{\mu_{\mathrm{F}}}$ coincides with the density of unpaired atoms $n_{\mathrm{F}} - n_{\mathrm{B}}$, as can be inferred from the expression (28) with $a_{\mathrm{BF}} = 0$, so that the expression (27), where the above self-energies appear in the combination $\Sigma_{\mathrm{CF}} - \Sigma^0_{\mathrm{F}}$ [51], is recovered. Inserting the full expression (28) for $\mu_{\mathrm{F}}$ in $\Sigma_{\mathrm{CF}}$, one obtains the next-order correction to (27)

$$\Sigma_{\mathrm{CF}} = \frac{4\pi a_{\mathrm{BF}}}{m_r}(n_{\mathrm{F}} - n_{\mathrm{B}})\left[1 + \frac{4}{3\pi}\frac{m_{\mathrm{F}}}{m_{\mathrm{r}}}\frac{x}{(1-x)^{2/3}}\frac{1}{g_{\mathrm{BF}}}\right]^{3/2} \tag{A.2}$$

$$\simeq \frac{4\pi a_{\mathrm{BF}}}{m_r}(n_{\mathrm{F}} - n_{\mathrm{B}})\left[1 + \frac{2}{\pi}\frac{m_{\mathrm{F}}}{m_{\mathrm{r}}}\frac{x}{(1-x)^{2/3}}\frac{1}{g_{\mathrm{BF}}}\right], \tag{A.3}$$

whereby in the brackets in the second line an expansion to first order in $1/g_{\mathrm{BF}}(\equiv k_{\mathrm{F}} a_{\mathrm{BF}})$ is carried out since $g_{\mathrm{BF}} \gg \frac{4}{3\pi}\frac{m_{\mathrm{F}}}{m_{\mathrm{r}}}\frac{x}{(1-x)^{2/3}}$ in the strong-coupling limit.

As a result, the strong-coupling expression (27) for $\mu_{\mathrm{B}}$ is improved as follows

$$\begin{aligned} \mu_{\mathrm{B}} = -\epsilon_0 &+ \frac{\left(6\pi^2 n_{\mathrm{B}}\right)^{2/3}}{2M} - \frac{\left[6\pi^2(n_{\mathrm{F}} - n_{\mathrm{B}})\right]^{2/3}}{2m_{\mathrm{F}}} - \frac{4\pi a_{\mathrm{BF}}}{m_{\mathrm{r}}} n_{\mathrm{B}} \\ &+ \frac{4\pi a_{\mathrm{BF}}}{m_{\mathrm{r}}}(n_{\mathrm{F}} - n_{\mathrm{B}})\left[1 + \frac{2}{\pi}\frac{m_{\mathrm{F}}}{m_{\mathrm{r}}}\frac{x}{(1-x)^{2/3}}\frac{1}{g_{\mathrm{BF}}}\right]. \end{aligned} \tag{A.4}$$

Note that the next-order correction to $\Sigma^0_{\mathrm{F}}$ (which enters both Eq. (27) for $\mu_{\mathrm{B}}$ and Eq. (28) for $\mu_{\mathrm{F}}$) is of order higher than two. As a consequence, the fermionic chemical potential does not have any second-order correction.

Differentiation of Eq. (A.4) with respect to $n_{\mathrm{B}}$ provides the second-order correction to $\mathcal{M}^{\mathrm{strong}}_{\mathrm{BB}}$, which now reads

$$\mathcal{M}^{\mathrm{strong}}_{\mathrm{BB}} = \frac{2\pi^2}{m_{\mathrm{F}} k_{\mathrm{F}}}\left(\frac{1}{(1-x)^{1/3}} + \frac{m_{\mathrm{F}}}{Mx^{1/3}} - \frac{4}{\pi}\frac{m_{\mathrm{F}}}{m_{\mathrm{r}}}\frac{1}{g_{\mathrm{BF}}} + \left(\frac{2}{\pi}\right)^2\left(\frac{m_{\mathrm{F}}}{m_r}\right)^2\frac{1-(4/3)x}{(1-x)^{2/3}}\frac{1}{g^2_{\mathrm{BF}}}\right). \tag{A.5}$$

The above second-order correction to $\mu_{\mathrm{B}}$ also affects the second-order expansion of $\mathcal{M}^{\mathrm{strong}}_{\mathrm{BF}}$. In this case, however, second-order corrections yield in practice a negligible contribution and are therefore not considered. Finally, by combining Eq. (A.5) for $\mathcal{M}^{\mathrm{strong}}_{\mathrm{BB}}$ with $\mathcal{M}^{\mathrm{strong}}_{\mathrm{BF}}$ and $\mathcal{M}^{\mathrm{strong}}_{\mathrm{FF}}$ from Eq. (39), one obtains the next-order correction to $\det \mathcal{M}^{\mathrm{strong}}$ which has been used in Fig. 5(b).

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
