# Peer review of "Mechanical stability of resonant Bose-Fermi mixtures"

_SciPost Physics, doi:SciPost Phys. 19, 039 (2025)_

## Round 1 · Referee Report · Anonymous (Referee 1) · 2025-5-26

Report

In the manuscript "Mechanical stability of resonant Bose-Fermi mixtures" by C.Gualerzi et al., the Authors investigate the stability of a Bose-Fermi mixture with tunable Bose-Fermi interactions at zero temperature.

This problem has already been theoretically addressed (as discussed in the manuscript introduction), but previous works considered various limitations, such as weak Bose-Fermi coupling strength, narrow Feshbach resonances, low boson concentration, the absence of a Bose-Einstein condensate, or negligible condensate depletion. The present work aims to overcome these limitations and extends the many-body diagrammatic approach developed by one of the Authors in Phys. Rev. A 91, 023603 (2015).

In particular, the Authors address the problem of the mechanical stability of the Bose-Fermi mixture using different theoretical approaches, such as a many-body diagrammatic method and fixed-node Quantum Monte Carlo calculations, supported by experimental observations. The bosons-fermion interaction has been treated using a T-matrix formalism. As for the bosons-bosons interaction, the Authors found that the Popov approximation is required to obtain results consistent with fixed-node Quantum Monte Carlo calculations.

After a detailed analysis of the formalism, the Authors investigate the mechanical stability of the mixture, showing that among the combinations currently accessible in experiments, the optimal mass ratio for achieving stability is that of the 87Rb-40K mixture. Moreover, by comparing their theoretical results with the outcomes of the 23Na-40K experiment reported in Nat. Phys. 19, 720 (2023), they conclude that the boson-boson repulsion in that case was too weak to ensure stability, and suggest that instability was not observed due to the short experimental time scale. Finally in the last part of the manuscript, the Authors explore the possibility and propose a strategy to observe the formation of a Bose-Einstein condensate by adiabatically reversing the BF pairing process, starting from the molecular phase, even in the absence of favourable BB and BF resonances.

In the Conclusions, the Authors provide a list of possible directions for further investigation, such as finite temperature effects, dynamical studies and the inclusion of higher-order diagrammatic corrections.

The manuscript is well written and presents original results which will have an important impact on the study of FB ultracold mixtures. Nowadays, various experiments investigate Bose-Fermi mixtures with different mass ratios and interaction strengths, making this study particularly timely and relevant. As clearly emphasized by the Authors in the introduction, Bose-Fermi mixtures are compelling systems for the quantum simulation of a wide range of physical phenomena.

For these reasons, I believe that the manuscript deserves publication on SciPost, provided that the following points are addressed by the Authors:

Eq. 5: it seems to me that the \Theta is not defined. If this is the case, I kindly ask the Authors to provide a clear definition.

End of Section 3.1, the Authors introduce Eqs. (29), (30), and (31) recasting the expressions for \mu_B and \mu_F and switching from a description in terms of a BF atomic mixture, to one based on an FF mixture composed of BF dimers and unpaired fermions. They introduce the dimer-fermion scattering length a_DF, and later state: “For equal masses (mB =mF) it yields 𝑎DF =8/3𝑎BF [86], to be compared with the exact value 𝑎DF =1.18𝑎BF [87,88]. I kindly ask the Authors to comment on this discrepancy. If they consider it a relevant aspect, the Authors may also comment on the effects of inhomogeneity introduced by a confining potential, especially in the case where it acts differently on the two species.

  1. Finally, I would be interested in a brief comment on the opposite regime nB>nF, which is not addressed in the present work.

Recommendation

Publish (meets expectations and criteria for this Journal)

  • validity: high
  • significance: high
  • originality: high
  • clarity: high
  • formatting: good
  • grammar: excellent

Author:  Leonardo Pisani  on 2025-06-25  [id 5600]

(in reply to Report 1 on 2025-05-26)

We thank the Referee for their positive report and for their suggestions to improve our manuscript. Below please find our response to all comments, questions, or suggestions raised by the Referee.

“Eq. 5: it seems to me that the \Theta is not defined. If this is the case, I kindly ask the Authors to provide a clear definition.”

In the revised version of our manuscript, we have clarified that \Theta indicates the Heaviside step function (see sentence just after Eq. (5)).

“End of Section 3.1, the Authors introduce Eqs. (29), (30), and (31) recasting the expressions for \mu_B and \mu_F and switching from a description in terms of a BF atomic mixture, to one based on an FF mixture composed of BF dimers and unpaired fermions. They introduce the dimer-fermion scattering length a_DF, and later state: “For equal masses (mB =mF) it yields a_DF =8/3a_BF [86], to be compared with the exact value a_DF =1.18a_BF [87,88]. I kindly ask the Authors to comment on this discrepancy.”

We have now explained the origin of this discrepancy in the last paragraph of Section 3.1, where we have added four new sentences (with two new references).

“If they consider it a relevant aspect, the Authors may also comment on the effects of inhomogeneity introduced by a confining potential, especially in the case where it acts differently on the two species.”

We have inserted a comment on this in the second to last paragraph of the introduction (with one additional reference)

“Finally, I would be interested in a brief comment on the opposite regime nB larger than nF, which is not addressed in the present work.”

We have added a comment on this point at the end of Section 4 (see the new paragraph preceding the last sentence of Section 4). This new comment also includes two new references.

---

## Round 1 · Referee Report · Anonymous (Referee 2) · 2025-6-3

Report

This paper investigates the competition between boson condensation and boson–fermion (BF) coupling in a BF mixture, at zero temperature, in which the boson density is smaller than the fermion density. The competition is controlled through a tunable Feshbach resonance. The mechanism of formation of BF moleculess is revisited highlighting its link with the depletion of the condensate fraction and the stabilizing effects of the fermionization process. In particular, in the absence of boson repulsion, BF mixtures are shown to feature an intrinsic stability regime whose occurrence precedes the transition to the phase characterized by the vanishing of the Bose condensate. The dependence of the mechanical stability of the mixture from the boson-repulsion parameter is explored for a wide interval of BF coupling by exploiting the determinant of the stability matrix while the derivation of the stability phase diagram of the system (this includes various significant cases at experimental level) allow one to evince the optimal-stability conditions. An experimental scheme is proposed to observe the critical phenomena of the mixture in the presence of mechanical stability.

The results presented in this paper are technically sound. The stability properties are found by applying well-established many-body diagrammatic approaches, the two versions of TMA procedure involving either the Bogoliubov or the Popov approximation. These are always compared with the results found with other approximation techniques or with the numerical methods applied in previous papers. A physical interpretation for the deviations emerging from the comparison is always supplied.

Except for section "Formalism", providing a very technical review of the theory on which this paper is based, the paper is, in general, well organized and well written. In particular, despite the focus on a rather specific field (the BF mixtures and their properties), I think it should be accessible to a broad audience.

The results of this paper feature an intrinsic interest at the theoretical level due to its thorough analysis both of the fermionization process and of the mechanical stability of mixtures which includes various realistic choices of the mixture parameters. I expect that the large amount of new information will stimulate further theoretical work and useful experimental applications.

Apart from some minor revision devoted to improve the readability of the paper in some unclear points (see the Request of changes below), this paper could be published in the current form.

Requested changes

1) Unlike the definition of parameter U_BB, the definition of parameter v^BF_0 (subsection 2.1) is not clear and does not allow to understand the link between v^BF_0 (parametrized in terms of a_BF) and the coupling strength g_BF describing the intensity of BF interaction. The latter is defined but does not appear in the model Hamiltonian. At present, both the role of symbol g_BF, and the behavior characterizing v^BF_0 in the strong and weak attraction regimes are not clear. I think that this point could be easily clarified.

2) Concerning figs. 6 (where white regions are stable) and 7, a short comment (in the captions or in the text) could provide a completely clear identification of unstable domains.

3) At page 16, the transition to the normal phase is associated to the critical value g^c_BF (third paragraph). Apparently, in the previous discussion, this value was related with the depletion of the condensate or, equivalently, with the formation of boson-fermion molecule and the fermionization process. A short comment should be introduced to clarify the presence of a residual fraction of boson in the normal-fluid phase.

Recommendation

Ask for minor revision

  • validity: top
  • significance: top
  • originality: high
  • clarity: top
  • formatting: excellent
  • grammar: excellent

Author:  Leonardo Pisani  on 2025-06-25  [id 5599]

(in reply to Report 2 on 2025-06-03)

We thank the Referee for their positive report and for their suggestions to improve our manuscript. Below please find our response to the three requested changes by the Referee.

“1) Unlike the definition of parameter U_BB, the definition of parameter v^BF_0 (subsection 2.1) is not clear and does not allow to understand the link between v^BF_0 (parametrized in terms of a_BF) and the coupling strength g_BF describing the intensity of BF interaction. The latter is defined but does not appear in the model Hamiltonian. At present, both the role of symbol g_BF, and the behavior characterizing v^BF_0 in the strong and weak attraction regimes are not clear. I think that this point could be easily clarified.”
Actually, the formal connection between v^BF_0 and a_BF (and thus g_BF) was given in section 2.2, Eq. (4). We now explicitly refer to that Equation in section 2.1. In addition, in section 2.2 we now give more details on the regularization procedure for the contact potential leading to Eq. (4). See new paragraph after Eq. (5) containing also two new equations (6) and (7). Finally, in both section 2.1 and 2.2 we have added an additional reference to a review paper for the regularization procedure, besides the two references we already quoted.

“2) Concerning figs. 6 (where white regions are stable) and 7, a short comment (in the captions or in the text) could provide a completely clear identification of unstable domains.”

We have added a short comment at the end of the captions of Fig. 6 and 7 to provide a clear identification of the unstable domains, as requested by the Referee.

"3) At page 16, the transition to the normal phase is associated to the critical value g^c_BF (third paragraph). Apparently, in the previous discussion, this value was related with the depletion of the condensate or, equivalently, with the formation of boson-fermion molecule and the fermionization process. A short comment should be introduced to clarify the presence of a residual fraction of boson in the normal-fluid phase."

We confirm that in our work we identify the fully fermionized phase with the absence of a bosonic condensate, assuming in practice that the depletion of the condensate is solely determined by the formation of molecules in the strong coupling region (g_{BF} larger than g^c_{BF}). In this way, we circumvent the problem of the blurred definition of the number of molecules (from which the number of unpaired non-condensed bosons would be obtained by difference from n_{B}) away from the strong-coupling limit g_{BF}\to+\infty. The point is that, in a many-body environment, molecules emerge as sharply defined entities only when they do not overlap at all, i.e. in the limit g_BF \to +\infty. For possible definitions (prompted by specific experimental protocols to identify molecules) in the analogous problem emerging in the study of the BCS-BEC crossover, see Refs [97] and [98].

We have added a footnote (at the end of section 3.3) to clarify this issue.

---

## Round 2 · Referee Report · Anonymous (Referee 2) · 2025-7-1

Strengths
2-the paper is, in general, well organized and well written. For this reason, despite the focus on a rather specific field (the BF mixtures and their properties) and the presence of some rather technical parts, I think it should be accessible to a broad audience.
Report
Requested changes
no further changes are requested
Recommendation
Publish (surpasses expectations and criteria for this Journal; among top 10%)
Strengths
- The presented results are original and will have an important impact on the study of Fermi-Bose ultracold mixtures
- The topic addressed is timely and relevant
Report
Requested changes
no further revisions are needed
Recommendation
Publish (easily meets expectations and criteria for this Journal; among top 50%)

---

## Round 2 · Author Response

Thank you for your Editorial Recommendation of June 17th, 2025.
As you recommended, we have considered all suggestions and requested changes by the Referees, and have implemented them in the revised version of our manuscript. Together with this resubmission, we have also sent a point-by-point response to both Referees and uploaded a list of changes.
We thank both Referees for their valuable comments and suggestions which helped us to improve our manuscript.
Yours sincerely,
Christian Gualerzi, Leonardo Pisani, Pierbiagio Pieri

---

## Round 2 · List of Changes

-Sec. 1, added new paragraph “In our calculations, we will focus on homogeneous mixtures…” (second to last paragraph of Sec. 1).
-Sec. 2.1, added “(see Eq. (4) below)” at the end of the second sentence of the first paragraph after Eq. (1).
-Sec. 2.2, added “and $\Theta(x)$ is the Heaviside step function of argument x .” in the first line after Eq. (5).
-Sec. 2.2, added new paragraph starting with “Equation (4) is obtained by ...” (first paragraph after Eq. (5)). This new paragraph contains two new equations (6) and (7).
-Sec. 3.1, added 4 new sentences (last four sentences of Sec. 3.1).
-Sec. 3.3, added footnote 1 at the end of Sec. 3.3.
-Bibliography, added Refs [59,61,77,78,81,85,95,97,98,101,102].
- Bibliography, eliminated former Ref. [26], which was an old preprint version of former Ref. [30], now Ref. [29].

---

## Editorial Decision

published